# Performance Evaluation of Mangrove Species Classification Based on Multi-Source Remote Sensing Data Using Extremely Randomized Trees in Fucheng Town, Leizhou City, Guangdong Province

**Xinzhe Wang [1], Linlin Tan [1] and Jianchao Fan [2,*]**

[1]  Institute of Information Science and Engineering, Dalian Polytechnic University, Dalian 116034, China
[2]  Department of Marine Remote Sensing, National Marine Environmental Monitoring Center, Dalian 116023, China
*   Correspondence: jcfan@nmemc.org.cn; Tel.: +86-411-8478-3328

**Abstract:** Mangroves are an important source of blue carbon that grow in coastal areas. The study of mangrove species distribution is the basis of carbon storage research. In this study, we explored the potential of combining optical (Gaofen-1, Sentinel-2, and Landsat-9) and fully polarized synthetic aperture radar data from different periods (Gaofen-3) to distinguish mangrove species in the Fucheng town of Leizhou, Guangdong Province. The Gaofen-1 data were fused with Sentinel-2 and Landsat-9 satellite data, respectively. The new data after fusion had both high spatial and spectral resolution. The backscattering coefficient and polarization decomposition parameters of the fully polarized SAR data which could characterize the canopy structure of mangroves were extracted. Ten different feature combinations were designed by combining the two types of data. The extremely randomized trees algorithm (ERT) was used to classify the species, and the optimal feature subset was selected by the feature selection algorithm on the basis of the ERT, and the importance of the features was sorted. Studies show the following: (1) When controlling a single variable, the higher the spatial resolution of the multi-spectral data, the higher the interspecific classification accuracy. (2) The coupled Sentinel-2 and Landsat-9 data with a 2 m resolution will have higher classification accuracy than a single data source. (3) The selected feature subset contains all types of features in the optical data and the polarization decomposition features of the SAR data from different periods: multi-spectral band > texture feature > polarization decomposition parameter > vegetation index. Among the optimized feature combinations, the classification accuracy of mangrove species was the highest, the overall classification accuracy was 90.13%, and Kappa was 0.84, indicating that multi-source and SAR data from different periods coupling could improve the discrimination of mangrove species. (4) The ERT classification algorithm is suitable for the study of mangrove species classification, and the classification accuracy of extremely random trees in this paper is higher than that of random forest (RF), K-nearest neighbor (KNN), and Bayesian (Bayes). The results can provide technical guidance and data support for mangrove species monitoring based on multi-source satellite data.

**Keywords:** mangrove community; integrated learning algorithm; full polarimetric SAR; polarimetric decomposition; multispectral

## 1. Introduction

Mangroves are a kind of salt-tolerant woody plant distributed along the tropical and subtropical coastlines. Mangrove wetland and surrounding organisms constitute an important mangrove wet-land ecosystem, which has great ecological value and economic benefits [1]. Mangroves play an important ecological role in protecting embankments, purifying water quality, carbon fixation and emission reduction, and maintaining biodiversity. At the same time, mangroves can also provide living materials and tourism resources.

However, in the past decades, due to the impact of reclamation, urbanization, ecological environment change, and other factors, the global mangroves have a trend of degradation [2]. The study of mangrove species range is crucial to the calculation of mangrove biomass carbon storage, which lays a research foundation for the conservation of mangrove ecosystem and vegetation community composition.

The habitat of mangroves is complex, and the changes of tidal ebb and flow in the intertidal area cause adverse effects on large-scale field surveys. Remote sensing technology can achieve the monitoring of mangrove forests in a large area. The data sources for inter-specific classification of mangroves using remote sensing data include two main categories: optical data, including multispectral and hyperspectral, and SAR data. The earliest low and medium resolution Landsat [3–6] were used for mangrove species identification. With the development of satellite technology, high spatial resolution IKONOS, Worldview-2, Worldview-3, Quick bird, Pléiades-1, and other satellite data were used for interspecific mangrove classification [7–10]. Because of the subtle differences in reflectance of different species of mangrove with different leaf structures, different mangrove species can be distinguished from each other using the specific band reflectance of optical data and its derived indices [11–13] and the distinct textural features of high-resolution satellites are beneficial to improve the interspecific classification accuracy of mangroves [8]. Hati et al. evaluated the potential of aerial hyperspectral AVIRIS-NG data in distinguishing mangrove species in Sundarbans Lothian, India [14]. The penetration ability of SAR data varies with different bands. The high frequency band has weak penetration ability, which only responds to the surface characteristics of mangrove canopy, while the low frequency band can penetrate the canopy to detect the trunk and ground, and respond to the features below the canopy. Therefore, SAR data are often used to monitor mangrove biomass [15], and a recent study by Ferrentinod et al. showed that the combination of SAR data at different frequencies can improve mangrove species identification performance.

SAR data can provide mangrove structure information complementary to optical data. Recently, some recent studies have used the combination of optical and SAR data to identify different mangrove species. Zhang et al. combined Worldview-3 and Radarsat-2 data to map the distribution of mangrove species in Mai Po Marshes Nature Reserve, Hong Kong, using a rotating forest classifier [16]. Kripa et al. monitored the current distribution of mangrove species in India using optical and SAR data [17]. Wong and Fung also found that the combination of hyperspectral data and SAR data could improve the interspecific classification accuracy of mangrove species [18]. A single optical satellite data source has the problem of low resolution or insufficient band information. In this study, Sentinel-2 and Landsat 9 satellite images are fused with GaoFen-1, respectively, to improve their spatial resolution for the first time, and the fused data and GaoFen-3 SAR data are used for mangrove classification research. The research on the application of the extreme random tree algorithm to mangrove species classification has not been reported, Therefore, the purpose of this paper is to study the use of the extreme random tree algorithm on the basis of multi-source data to evaluate the differentiation potential of different features on mangrove species(Avicennia marina (AM), Kandelia obovate (KO) and Sonneratia apetala (SA)) and further use the recursive feature elimination algorithm to screen the optimal feature combination so as to remove some redundant features. In order to verify the effectiveness of the extreme random tree algorithm, random forest, KNN, and Bayes algorithms are selected on the basis of the filtered features for comparison. The data fusion method and classification algorithm in this paper can provide new ideas and technical references for mangrove species monitoring.

## 2. Materials and Methods

### 2.1. Study Area

The research area is located in the coastal area of Fucheng Town on the east coast of Leizhou Peninsula, Guangdong Province, China, with a geographic location of 110°9′37.62″E~110°10′10.61″E, 20°54′46.12″N~20°57′46.41″N. The detailed geographic location is shown

in Figure 1. The total area of mangroves in Guangdong Province is about 12,039.80 ha, which is mainly distributed in western Guangdong and Leizhou Peninsula. The area of mangroves in Leizhou Peninsula is 7905.56 ha [18]. The area of mangroves in Leizhou Peninsula has increased in recent years. The region has a subtropical monsoon climate, with an average annual temperature of about 22.3 °C [19] and an average annual precipitation of about 1500 mm. The tide on the east coast of Leizhou Peninsula is irregular semidiurnal tide with an average tidal range of 3.12 m [20]. The main mangrove species in the region are *Avicennia marina* (AM), *Kandelia obovata* (KO) and *Sonneratia apetala* (SA). In 1995, the species of SA was artificially introduced into the town. Up to now, the diameter at breast height of the plantation is 13–20 cm, and the plant height is about 12 cm. The natural regeneration pattern of the SA community in the town is clustered and distributed. The average height of SA is significantly higher than that of KO and AM. The mixed degree of mangroves in the town is at a moderate level, and the community structure is simple. The artificial forest of SA is mainly distributed in the high tide zone and the middle tide zone, while AM is mainly distributed in the low tide zone, and KO is more suitable for the environment of the middle high tide zone. At the same time, due to the sparse and scattered distribution and small number of *Rhodophora stylosa* and *Bruguiera gymnorrhiza* [21], remote sensing does not image them, and the impact on the ecological environment can be ignored, which is of no research value for this paper as the classification objective of this paper is to identify the three dominant species of AM, KO and SA.

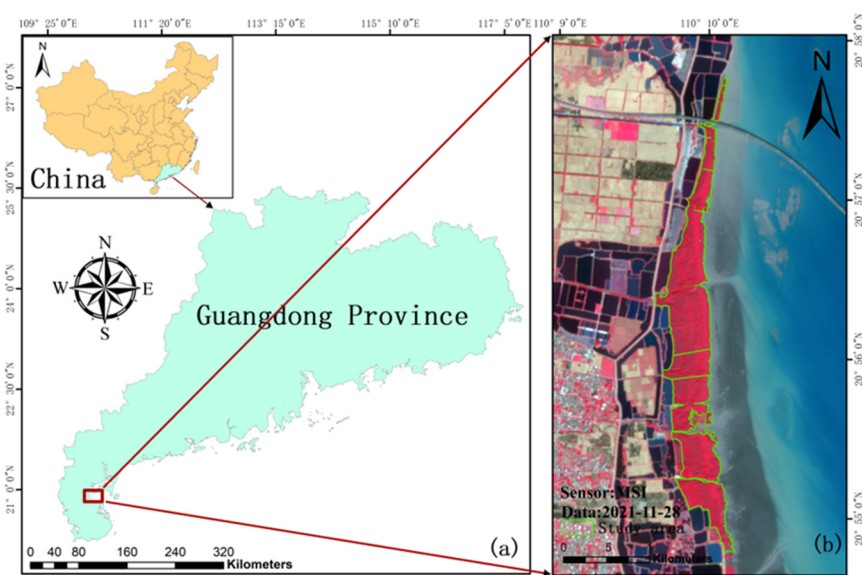

**Figure 1.** Location of the study area: (**a**) location of the mangrove forest in Fuchengtown of Leizhou City, Zhanjiang City, Guangdong Province (map taken from the standard map service website http://bzdt.ch.mnr.gov.cn (accessed on 15 December 2022), review number GS (2019)1822); (**b**) false color composite map of 8, 4, and 3 bands of Sentinel-2 satellite data in the study area.

*2.2. Field Data*

This study went to the study area for field investigation in the middle of May 2022. A total of 15 sample points including the three species of SA, KO, and AM were collected. These field samples were randomly sampled by handheld GPS along the inner and outer edges of mangroves and the tidal gullies, with a quadrate size of 10 m$^2$. The species name and sample point location were recorded. Then the hue and texture characteristics of each species will be analyzed in combination with the high-resolution remote sensing images. The interpretation marks of different mangrove species are shown in Table 1.According to the field investigation and literature research, the samples will be expanded to 150 points through visual interpretation.

**Table 1.** Interpretation marks of different mangrove species.

| Species Name | Field Pictures | Remote Sensing Image | Interpretation Features |
|---|---|---|---|
| Avicennia marina | | | Avicennia marina is bright red in the remote sensing image, and the texture is smooth and fine |
| Kandelia obovata | | | The Kandelia obovata is dark red in the remote sensing image, with fine texture |
| Sonneratia apetala | | | The Sonneratia apetala is dark red in the remote sensing image, with rough texture |

*2.3. Remote Sensing Data and Pre-Processing*

2.3.1. Optical Data and Pre-Processing

The optical data used in this paper include the images of Gaofen-1, Sentinel-2, and Landsat-9 satellites. Gaofen-1 is the first high-resolution earth observation satellite launched by China on 26 April 2013, which has a spatial resolution of 8 m in the multispectral band and 2 m in the panchromatic band. The multispectral band can provide four bands of blue, green, red, and near infrared. ESA's Sentinel-2 data, launched in March 2017, can cover 13 spectral bands with varying spatial resolution from the visible and near-infrared to the short-wave infrared. Among the optical data, Sentinel-2 data is the only one that contains three bands in the red-edge range. Sentinel-2 data is available from https://scihub.copernicus.eu/dhus/#/home, accessed on 11 May 2022, website for free download. This paper uses Landsat-9 satellite data in 2022. The bands used in the study include 7 multispectral bands and 1 panchromatic band. The Landsat data is available on the website of the United States Geological Survey https://earthexplorer.usgs.gov/Download, accessed on 14 May 2022. This paper only preprocessed the panchromatic band of Gaofen-1, including radiometric calibration and orthorectification correction. All the above operations are completed in ENVI5.3 software. Sentinel-2 has already performed radiometric calibration and atmospheric correction before downloading data, so it only needs to use SNAP software to resample to 10 m and then fuse the processed high Gaofen-1 data with Sentinel-2 data. The fusion algorithm is Gram Schmidt Pan Sharpening. The resolution of Sentinel 2 data after fusion was 2 m. The pre-processing steps of the Landsat-9 data include radiation calibration, atmospheric correction, and fusion. The resolution of the fused data was 15 m. The Landsat-9 data were also fused with the processed Gaofen-1 data. The fusion algorithm was the same as above, and the resolution after fusion was also 2 m.

2.3.2. SAR Data Pre-Processing

Gaofen-3 satellite is the first Chinese C-band multi-polarization SAR satellite launched by the National Space Administration of China in August 2016.Two views of Gaofen-3

full polarization data were selected for this study: one view was collected on 5 February 2021 and the other on 13 May 2022. Since there was basically no change in this region from 2021 to 2022, the impact of date on interspecific classification can be ignored, including four polarization modes (HH, HV, VH, VV). The detailed parameters of the data were shown in Table 2. The Gaofen-3 data used in the study were obtained from the China Ocean Satellite Data Service (http://osdds.nsoas.org.cn/, accessed on 10 June 2022). The Gaofen-3 data were preprocessed using the SARscape 5.6 software embedded in ENVI 5.6. The preprocessing steps included multi-view processing (obtaining intensity data from complex intensity data), single-channel intensity data filtering (to remove speckle noise), geocoding and radiometric calibration (to obtain the backscatter coefficients of the data). After registration with optical data, the resolution of the radar image was 8 m. In order to match the resolution of the optical data, resampling to 2 m was also required.

**Table 2.** Remote sensing data parameters.

| Satellite | Date | Bands | Spatial Resolution (m) |
|---|---|---|---|
| GaoFen-1 | 15 May 2021 | 4 | 2 |
| GaoFen-3 | 2 February 2021 | 4 | 8 |
| GaoFen-3 | 13 May 2022 | 4 | 8 |
| Sentinel-2 | 28 November 2021 | 12 | 10 |
| Landsat-9 | 4 May 2022 | 7 | 15 |

*2.4. Optical and SAR Data Conversion*

In order to improve the degree of differentiation of the spectral features of mangrove species, seven vegetation index features derived from the fused Sentinel-2 data and two new band combination features are used as inputs in addition to the original reflectance of the spectral data. In this paper, texture features of mangroves are extracted from optical images based on the grayscale co-occurrence matrix, and the higher the resolution of the images, the richer the spatial detail information provided by the texture features; thus, the texture information of mangroves is extracted using the panchromatic images of Gaofen-1. The specific spectral index features and texture features are shown in Table 3. Because this paper focuses on the complementarity between the polarization features of SAR images and optical images and the potential of SAR images to distinguish mangrove species, it does not specifically discuss the influence of the gray level co-occurrence matrix parameters on this study. In this paper, a 3 × 3 window is used to calculate the gray level co-occurrence matrix. A total of seven texture parameters are selected, including mean, variance, homogeneity, contrast, dissimilarity, entropy, and angular second moment (ASM).

Microwave radiation in the C-band tends to interact most actively with the leaves and branches of the upper canopy [22], and identification using SAR requires differences in the structure (e.g., size, geometry) and dielectric constants of tree components between species [23]. Different species have different heights and canopy structures. Different characteristics may lead to different scattering mechanisms. Components of different scattering types can be extracted through various polarization decomposition methods, including volume scattering, surface scattering, and double-bounce scattering. It has been shown that H/A/α decomposition and Freeman decomposition can be used for intra-class identification of mangroves [24,25]. In this paper, PolSARpro radar data processing software was used to extract six polarization decomposition features of fully polarized Gaofen-3 data, which are three eigenvalues based on H/A/α decomposition, volume scattering, surface scattering, and double-bounce scattering component based on Freeman decomposition. The specific decomposition process is described below:

The target data obtained from fully polarized SAR images is the complex scattering matrix $S$:

$$S = \begin{bmatrix} S_{hh} & S_{hv} \\ S_{vh} & S_{vr} \end{bmatrix} \tag{1}$$

The scattering matrix is vectorized by Pauli basis to obtain the *k* matrix:

$$k = \frac{1}{\sqrt{2}} [S_{hh} + S_{vv} S_{hh} - S_{vv} 2S_{hv}] \tag{2}$$

Most ground objects are randomly distributed targets with multiple scattering centers. The total signal can be obtained by superposition of multiple coherent signals, which is the coherent interference of smooth electromagnetic waves. Therefore, the polarization covariance matrix *C* and the polarization coherence matrix *T* can be obtained by using the spatial statistical average processing. The two can be mutually transformed through the unitary matrix. The expression of the polarization coherence matrix is

$$T = \langle k \cdot k^{\mathrm{H}} \rangle = \frac{1}{N} \sum_{i=1}^{N} k_i \cdot k_i^{\mathrm{H}} \tag{3}$$

It should be noted that *N* is the apparent number, $k_i$ is the apparent scattering vector, the superscript H represents the conjugate transpose, and the coherence matrix *T* is the $3 \times 3$ semi-positive definite Hermite matrix. According to the matrix decomposition theory, the matrix *T* can be diagonalized and has three non-negative real eigenvalues, and the matrix *T* can be decomposed into the sum of three independent coherence matrices:

$$T = U \sum U^{-1} = \sum_{i=1}^{3} \lambda_i T_i = \sum_{i=1}^{3} \lambda_i u_i \cdot u_i^{\mathrm{H}} \tag{4}$$

$T_i$ represents the independent coherence matrix with rank 1; $\lambda_i$ is the real eigenvalue; $u_i$ is the normalized eigenvector corresponding to the eigenvalue; $i$ = 1, 2, 3 represent three scattering mechanisms. The eigenvalue $\lambda_i$ corresponds to the intensity of the scattering mechanism. Freeman decomposition is to decompose the polarization covariance matrix into three main scattering mechanisms, and the covariance matrix is expressed as

$$\langle C_3 \rangle = f_s C_{3s} + f_d C_{3d} + f_v C_{3v}$$

$C_{3s}, C_{3d}, C_{3v}$ are surface scattering, secondary surface scattering, and volume scattering models, respectively; $f_v, f_s, f_d$ are the weights of volume scattering, surface scattering, and double-bounce scattering, respectively. The scattering power of volume scattering, surface scattering, and double-bounce scattering are $P_v, P_s, P_d$, and $\alpha$ and $\beta$ are the parameters.

$$P_v = \frac{8 f_v}{3} \tag{5}$$

$$P_s = f_s \left( 1 + |\beta|^2 \right) \tag{6}$$

$$P_d = f_d \left( 1 + |\alpha|^2 \right) \tag{7}$$

**Table 3.** Sentinel 2 data-derived features.

| Variables | Definition | Source |
|:---:|:---:|:---:|
| DVI | $B8 - B4$ | [26] |
| MSAVI | $\frac{(B8-B4)*1.5}{B8+B4+0.5}$ | [23] |
| GNDVI | $\frac{B8-B3}{B8+B3}$ | [27] |
| IRECI | $\frac{B7-B4}{B5/B6}$ | [28] |
| SAVI | $1.5 \frac{B8-B4}{B8+2.4B4+0.5}$ | [29] |

**Table 3.** *Cont.*

| Variables | Definition | Source |
|---|---|---|
| RVI | $\frac{B8}{B4}$ | [30] |
| NDI45 | $\frac{B5-B4}{35+B4}$ | [31] |
| Mean | $\sum_i \sum_j p(i,j)*i$ | |
| Variance | $\sum_i \sum_j p(i,j)*(i-\text{Mean})^2$ | |
| Homogeneity | $\sum_i \sum_j p(i,j)*\frac{1}{1+(i-j)^2}$ | |
| Contrast | $\sum_i \sum_j p(i,j)*(i-j)^2$ | [32] |
| Dissimilarity | $\sum_i \sum_j p(i,j)*|i-j|$ | |
| Entropy | $\sum_i \sum_j p(i,j)*\ln p(i,j)$ | |
| ASM | $\sum_i \sum_j p(i,j)^2$ | |
| New band combination 1 | $B8 + B4$ | |
| New band combination 2 | $5(B9 - B4)$ | |

Note: $B3$ = Green, $B4$ = Red, $B5$ = Vegetation Red Edge 1, $B6$ = Vegetation Red Edge 2, $B7$ = Vegetation Red Edge 3, $B8$ = NIR, $B9$ = Narrow NIR, $p(i,j)$ is defined as the probability of leaving a position relation from a point with gray scale $i$ to a gray scale $j$.

### 2.5. Mangroves and Non-Mangroves: Decision Trees

The decision tree classification was used to extract mangrove spatial distribution information. The mangrove vegetation index (MVI) proposed by Baloloy et al. has been widely used for mangrove information extraction [33]. In this paper, MVI and the normalized vegetation index (NDVI) are used as thresholds of decision tree segmentation nodes. The threshold range of decision tree nodes is debugged according to the actual measured point mangrove samples, and the threshold value of each node is determined by debugging (Figure 2).

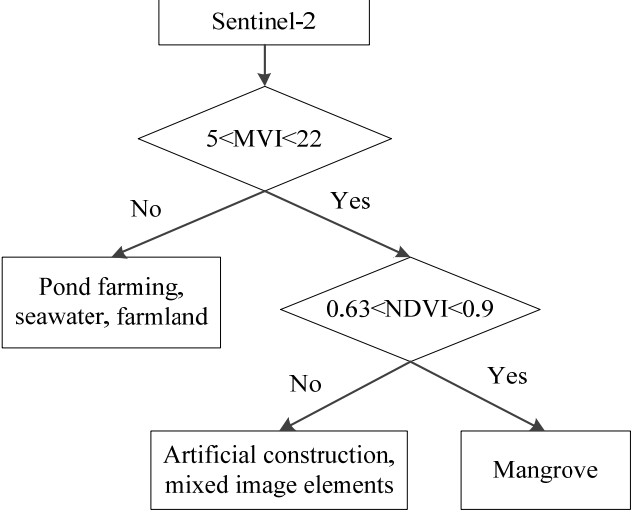

**Figure 2.** Decision tree model for identifying mangroves.

First of all, the first rule is 5 < MVI < 22. This rule divides the ground objects into two categories: one category is pond farming, sea water, and farmland, and the other category is artificial buildings, mixed pixels of pond boundary vegetation and water, and

mangroves. The second rule is 0.63 < NDVI < 0.9; the ground objects meeting this rule are mangroves, otherwise they are artificial buildings and mixed pixels. This rule can effectively distinguish mangroves from artificial buildings and mixed pixels at the pond boundary. After verification and calculation, the overall accuracy of mangrove extraction is 92.30%. The total area of mangrove extraction is 165.79 hectares.

*2.6. Mangrove Interspecies Classification: Extremely Randomized Trees*

Extremely randomized trees (ERT) is an integrated learning algorithm based on decision trees proposed by Pierre Geurts et al. ERT is a variant of random forest (RF). ERT directly uses all samples to randomly select $N$ features. For these $N$ features, each feature randomly selects one split node, thus obtaining $N$ split nodes. Then, ERT calculates the scores of the $N$ split nodes and selects the node with the highest score as the split node. Therefore, there are two main differences between the ERT algorithm and the RF algorithm: one is that for the training set of each decision tree, RF uses random sampling bootstrap to select the sample set as the training set of each decision tree, while ERT does not use random sampling, that is, each decision tree uses the original training set; second, after selecting the division features, RF's decision tree will select an optimal feature value division point based on the Gini coefficient, mean squared deviation and other principles, which is the same as the traditional decision tree. ERT will randomly select a feature value to divide and obtain a result with smaller and more stable variance compared to random forest. So far, ERT algorithms have been used for predicting surface latent heat fluxes, air quality and lake surface water temperature simulations, and mangrove range extraction [34–37]. However, its performance in mangrove species classification has not been explored. The specific steps of extreme random trees are as follows:

Input: training sample D; number of decision trees M; number of characteristics K.

Step 1: Each decision tree-based classifier uses all of sample D for training.

Step 2: Generate M decision trees based on CART algorithm. The process of splitting nodes is to randomly select k features from all K features, randomly select samples with some categories as left branches, and divide samples with other categories into split samples and repeat this process until the decision tree stops growing.

Step 3: Repeat steps 1 and 2 M times to finally form an extreme random tree classification model composed of M decision trees.

Step 4: Test the data set: Each decision tree of the extreme random tree will produce a classification result. The category with the most voting results is the final classification result.

Based on Python language, this paper uses the GDAL library to read raster data, calls the extreme random tree classifier of the integrated learning module of sklearn Library, sets the number of submodels to 500, keeps the default values of other parameters, and uses the overall accuracy and Kappa coefficient to evaluate the classification accuracy. Vector format samples are generated through the ArcGIS software, and the sample data are randomly split into training sets and test sets using Python, with the split ratio of 3 to 7, of which 30% is used as training samples and 70% as test samples. The classification method process of mangrove species is shown in Figure 3.

In order to verify the applicability of the extreme random tree algorithm, this paper selects another three commonly used machine learning algorithms for comparison, namely, Random forest (RF), K-nearest neighbor (KNN), and Bayesian (Bayes). RF is an integrated learning method that uses decision trees as weak learners. By building multiple decision tree models on data, integrating the modeling results of all decision tree models, and adopting majority voting, more appropriate results can be obtained. Therefore, the stochastic forest model has strong generalization ability and is not easy to overfit. RF is a very representative Bagging algorithm. Each decision tree uses a bootstrap method to randomly generate independently distributed training data and randomly select some eigenvalues to train the decision tree. Some studies have shown that 500 is a suitable value [38] when using RF classifiers on remote sensing data. Therefore, this paper sets the number of sub-models

to 500, and sets the maximum number of features to participate in the judgment during node splitting as the default value, that is, the square root of the number of input features. The core idea of the KNN algorithm is that the class of unlabeled samples is determined by the nearest *K* neighbors. When mangrove species are classified by the KNN algorithm, the *K* value is set to 5. The principle of Bayes classification is to statistically calculate the prior probability of training samples and then calculate a posteriori probability according to Bayes theorem to predict the category of the sample to be classified.

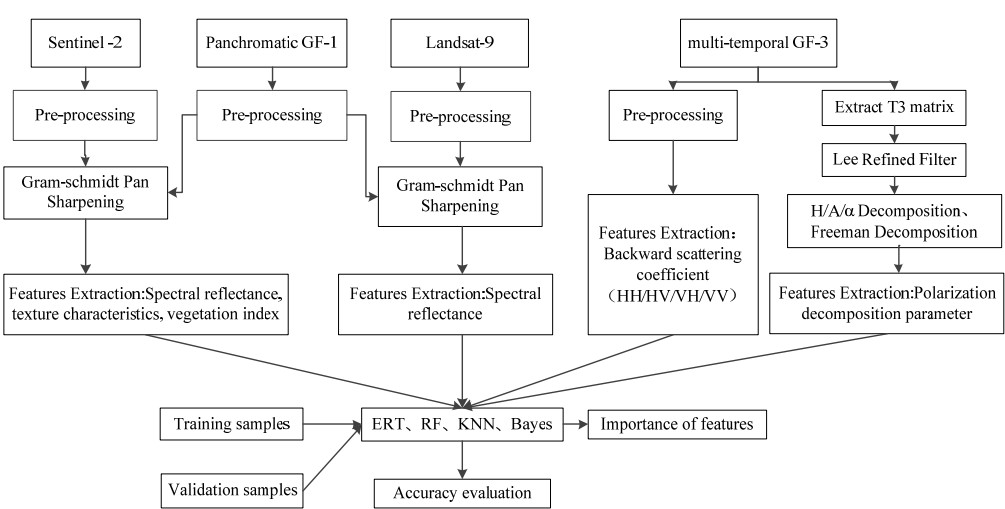

**Figure 3.** Methodological framework for mangrove species identification.

## 3. Results

### 3.1. Species Classification Results from Different Data Sources

A total of 10 sets of experiments were designed in this paper, and the data feature combinations and spatial resolutions of each set of experiments are shown in Table 4. The classification results are shown in Figure 4. The data sources of the first six sets of experiments were all optical data; the data sources of seven, eight, and nine sets of experiments were a combination of optical and radar data; and the data source of the last set of experiments was a subset of the preferred features, and the preferred features will be described in detail in the next section. The effect of multiple factors on the interspecies classification accuracy is investigated using the ERT algorithm, including factors such as the spatial resolution of multispectral data, the combination of optical data and fully polarized SAR data relative to a single type of data, etc. The classification accuracy table of 10 groups of experiments is shown in Table 5. Groups 1 and 2 are a set of control experiments: the first experimental input features are the original 12 bands of Sentinel 2 data with 10 m resolution, and the second group of experimental inputs were 12 bands of Sentinel 2 multispectral and Gaofen-1 panchromatic data fused with 2 m resolution; the species classification accuracy of Sentinel-2 data at 10 m resolution is 66.59%, and the classification accuracy of Sentinel-2 data at 2 m resolution after fusion is 84.91%. Groups 3 and 4 are a set of control experiments: the third group's experimental inputs featured the original seven bands of Landsat-9 at 15 m resolution, and the fourth group of experimental inputs is seven bands of Landsat-9 multispectral and 2 m resolution after the fusion of Gaofen-1 panchromatic data; the classification accuracy of Landsat-9 with 15 m resolution is 59.68%, and the classification accuracy of Landsat-9 data with 2 m resolution after the fusion is 80.45%. The two groups of control experiments show that the multispectral data with high spatial resolution will improve the mangrove species identification The input features of group 5 are the combination of all the features of groups 2 and 4; the classification accuracy is 85.49%. The input features of group 6 are the vegetation index, band combination and texture features added to the features of group 5, and the classification accuracy is 85.10%, which is slightly lower than that of group 5.

**Table 4.** Feature combination scheme, Sentinel-2, Landsat-9, and Gaofen-3 data and derived feature parameters. The polarization decomposition parameters include λ1, λ2, λ3 eigenvalues and the volume scattering, surface scattering and double-bounce scattering components of Freeman decomposition (2λ3 represents the λ3 eigenvalue in February; 5 Pv represents the volume scattering component in May).

| Serial No. | Data Used | Bands/Indices Used | Resolution (m) |
|---|---|---|---|
| 1 | Sentinel-2 | original 12 bands | 10 |
| 2 | Sentinel-2, Gaofen-1 | 12 bands of Sentinel-2 after fusion | 2 |
| 3 | Landsat-9 | original 7 bands | 15 |
| 4 | Landsat-9, Gaofen-1 | 7 bands of Landsat-9 after fusion | 2 |
| 5 | Sentinel-2, Landsat-9 Gaofen-1 | 12 bands of Sentinel-2 after fusion, 7 bands of Landsat-9 after fusion | 2 |
| 6 | Sentinel-2, Landsat-9 Gaofen-1 | 12 bands of Sentinel-2 after fusion, 7 bands of Landsat-9 after fusion, 9 indices, 7 texture features | 2 |
| 7 | Sentinel-2, Landsat-9 Gaofen-1, Gaofen-3 | 12 bands of Sentinel-2 after fusion, 7 bands of Landsat-9 after fusion, 9 indices, 7 texture features, backscattering coefficients HH, HV, VH, and VV, and polarization decomposition characteristics in February | 2 |
| 8 | Sentinel-2, Landsat-9 Gaofen-1, Gaofen-3 | 12 bands of Sentinel-2 after fusion, 7 bands of Landsat-9 after fusion, 9 indices, 7 texture features, backscattering coefficients HH, HV, VH, and VV, and polarization decomposition characteristics in May | 2 |
| 9 | Sentinel-2, Landsat-9 Gaofen-1, Gaofen-3 | 12 bands of Sentinel-2 after fusion, 7 bands of Landsat-9 after fusion, 9 indices, 7 texture features, backscattering coefficients HH, HV, VH, and VV, and polarization decomposition characteristics in February and May | 2 |
| 10 | Preferred subset of features | Sentinel-2 deep blue band, blue band, green band, DVI, 5 (b9-b4), Landsad-9's green, NIR, shortwave IR1, shortwave IR2 band, mean, GNDVI, IRECI, SAVI, 2λ3, 5 Pv | 2 |

Groups 6, 7 and 8 are a set of control experiments. The input of group 6 is all the multi-spectral data features with classification accuracy of 85.10%. The input of group 7 is based on the input features of group 6, adding the features of the fully polarized SAR data in February. The characteristics of fully polarized SAR data include the backscattering coefficient HH\HV\VH\VV, the three eigenvalues λ1\λ2\λ3 of the coherence matrix T, and the three components of Freeman decomposition. The classification accuracy of the seventh group is 87.95%, and the eighth group is the addition of the characteristics of the fully polarized SAR data in May on the basis of the input characteristics of the sixth group. The classification accuracy was 88.51%. When the backscattering coefficient and polarization decomposition feature of SAR data in February were added, the classification accuracy was increased by 2.85% compared with only optical features, and when the backscattering coefficient and polarization decomposition feature of SAR data in May were added, the classification accuracy was increased by 3.41% compared with only optical data. Group 9 input comprised 55 features, including optical features and SAR features of February and May, with a classification accuracy of 90%. Group 10 input comprised an optimal features subset with a classification accuracy of 90.13%, which shows that the accuracy of optimal features is slightly higher than that when all features are inputted. It shows that only using the preferred feature as the feature subset can achieve higher classification accuracy.

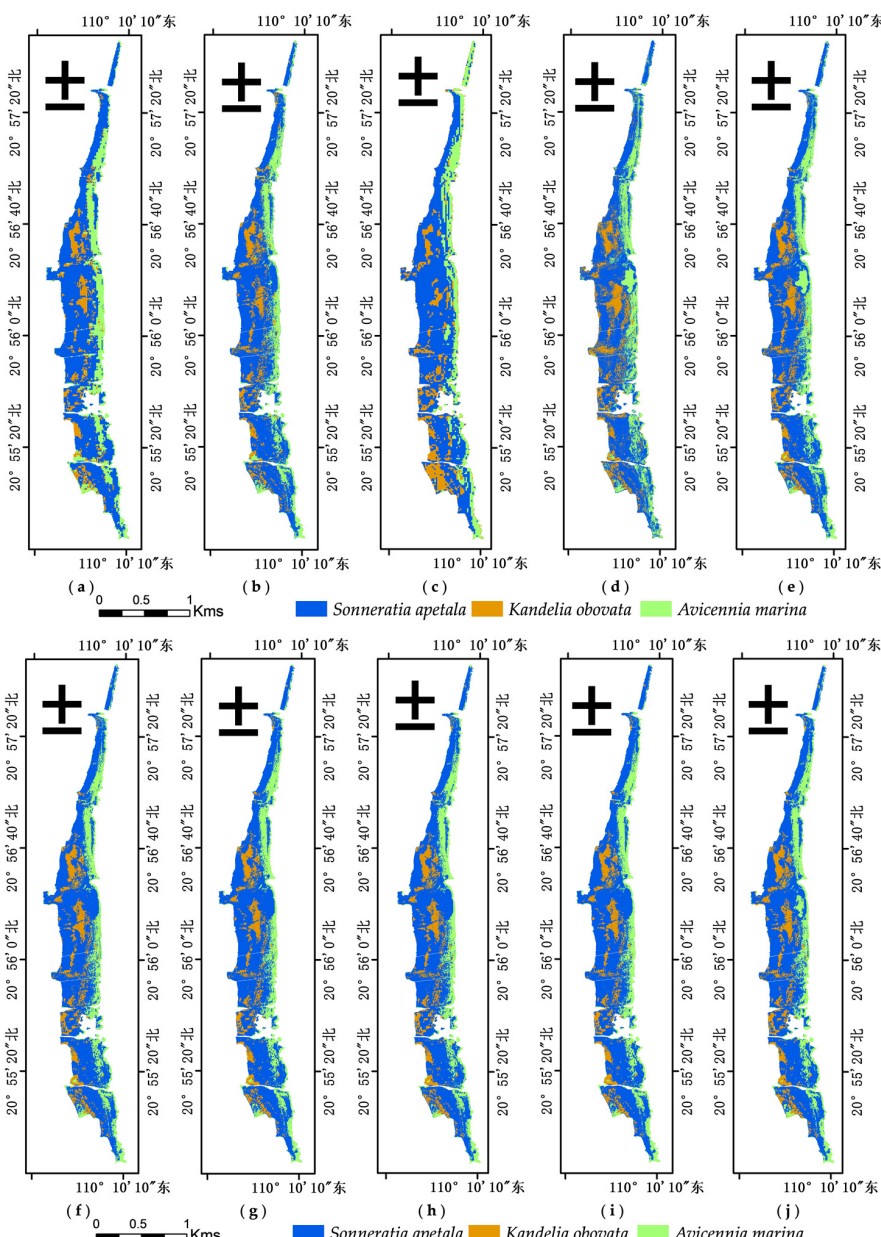

**Figure 4.** Mangrove species classification result maps based on ERT: (**a**–**j**) represent the classification result maps of 10 different sets of features; the order of input feature sets is consistent with the description in Table 3.

**Table 5.** Classification accuracy of mangrove species by ERT classification model.

| Serial No. | | ERT | | | OA | Kappa |
|---|---|---|---|---|---|---|
| | | SA | KO | AM | | |
| 1 | PA | 0.69 | 0.69 | 0.59 | 66.59% | 0.45 |
| | UA | 0.76 | 0.55 | 0.6 | | |
| 2 | PA | 0.84 | 0.86 | 0.85 | 84.91% | 0.75 |
| | UA | 0.89 | 0.78 | 0.84 | | |
| 3 | PA | 0.6 | 0.55 | 0.63 | 59.68% | 0.34 |
| | UA | 0.66 | 0.47 | 0.61 | | |

**Table 5.** *Cont.*

| Serial No. | | | ERT | | OA | Kappa |
|---|---|---|---|---|---|---|
| | | SA | KO | AM | | |
| 4 | PA | 0.81 | 0.79 | 0.8 | 80.45% | 0.68 |
| | UA | 0.84 | 0.75 | 0.8 | | |
| 5 | PA | 0.85 | 0.86 | 0.86 | 85.49% | 0.76 |
| | UA | 0.88 | 0.8 | 0.86 | | |
| 6 | PA | 0.84 | 0.86 | 0.87 | 85.10% | 0.75 |
| | UA | 0.9 | 0.77 | 0.83 | | |
| 7 | PA | 0.86 | 0.9 | 0.9 | 87.95% | 0.8 |
| | UA | 0.93 | 0.81 | 0.86 | | |
| 8 | PA | 0.86 | 0.9 | 0.92 | 88.51% | 0.81 |
| | UA | 0.93 | 0.81 | 0.86 | | |
| 9 | PA | 0.88 | 0.92 | 0.93 | 90% | 0.83 |
| | UA | 0.94 | 0.83 | 0.88 | | |
| 10 | PA | 0.89 | 0.91 | 0.91 | 90.13% | 0.84 |
| | UA | 0.93 | 0.86 | 0.90 | | |

Most of the mangroves in Fucheng town are artificial forests, so the mangrove community structure is relatively simple. The extraction results showed that the area of SA was 106.93 ha, and its species distribution was the most extensive, accounting for 64.49% of the total mangrove area in the study area. The distribution area of AM was 35.44 ha, accounting for 21.37% of the total mangrove area in the study area. The distribution area of KO was 23.42 ha, accounting for only 14.12%, scattered around the mid-tide zone and near the dike.

*3.2. User Accuracy and Producer Accuracy*

User accuracy means that a random sample is taken from the classification results, and the type it contains is the same as the actual type of the ground. Low user accuracy indicates that the target ground object is easy to be divided into other categories. Cartographic accuracy refers to the conditional probability that the classification result of the same location on the classification map is consistent compared to any random sample in the real data. Low cartographic accuracy indicates that some other ground objects are easy to be classified into target ground objects. When both producer accuracy and user accuracy are greater than 85%, the classification of such species is considered reliable [39]. In the first and third experiments, the producer accuracy and user accuracy of the three mangrove species were all lower than 75%, indicating that the remote sensing data of medium and low resolution had serious unreliability for interspecific classification. In the second experiment, the producer accuracy of the three species was all about 85%, and the User accuracy of KO was significantly lower than that of the other two species, indicating that the spectral band of Sentinel-2 could easily classify KO into the other two species. The user accuracy and producer accuracy of group 4 were both lower than 80%, indicating that there were more mixed pixels between KO and other two ground objects, and the spectral values of KO and other two species overlap more in the Landsat-9 data. In the fifth and sixth groups of experiments, the user accuracy of KO is significantly lower than that of the other two species, which indicates that the use of optical data alone cannot better distinguish KO from other species. The user accuracy and producer accuracy of the three species after the introduction of SAR data features in the seventh and eighth groups of experiments are higher than that of the optical data in the sixth group, indicating that SAR data can compensate for the optical spectrum overlap of optical data and improve the classification

accuracy of species. The producer accuracy of the three species in the ninth group was more than 88%, but the user accuracy of KO is 83% lower than that of the other two species. The user accuracy of the tenth group of experiments was 0.03% higher than that of the ninth group, indicating that the discrimination of KO could be improved by using the optimal features.

### 3.3. Data Feature Selection Results and Feature Importance

Feature redundancy occurs when the input feature dimension is too high. In this paper, we use the feature importance property of the ERT algorithm to calculate the feature importance of multispectral and fully polarized SAR features, rank the importance from high to low, and use the recursive feature elimination idea to filter the best combination of features, i.e., remove one feature with the lowest feature importance from the current feature subset each time to obtain a new feature subset. The importance of each feature in the new feature subset is calculated, and the selected features are involved in ERT classification each time; the overall accuracy and Kappa coefficient are calculated, and the relationship between the number of feature variables and the classification accuracy and Kappa coefficient is plotted to find the best combination of feature parameters and the best number of features. As can be seen from Figure 5a, when the number of features increases to a certain value, the overall classification accuracy and Kappa coefficient remain flat, as does the number of features. Through experiments, it is found that when the number of features is 16, the overall accuracy and Kappa coefficient both reach the maximum, which are 90.13% and 0.84, respectively, indicating that the optimal number of features is 16. It includes four types of features: original features of spectral data, vegetation index, texture feature and polarization decomposition parameter. The original features of spectral data include three wavebands with the central wavelengths of 443 nm, 490 nm and 560 nm for Sentinel-2 data and five wavebands with the central wavebands of 482.6 nm, 561.3 nm, 864.6 nm, 1609 nm and 2201 nm for Landsat-9 data. The vegetation index features contain six, respectively DVI, GNDVI, IRECI, SAVI, DVI, and 5 (b9-b4). The texture features contain only one mean value. The polarization decomposition parameters include the third eigenvalue of the coherence matrix of February SAR data and the volume scattering eigenvalue of May SAR data. The set of features contains optical data and SAR data from different periods, indicating that the interspecies classification accuracy can only be improved by combining multi-source data.

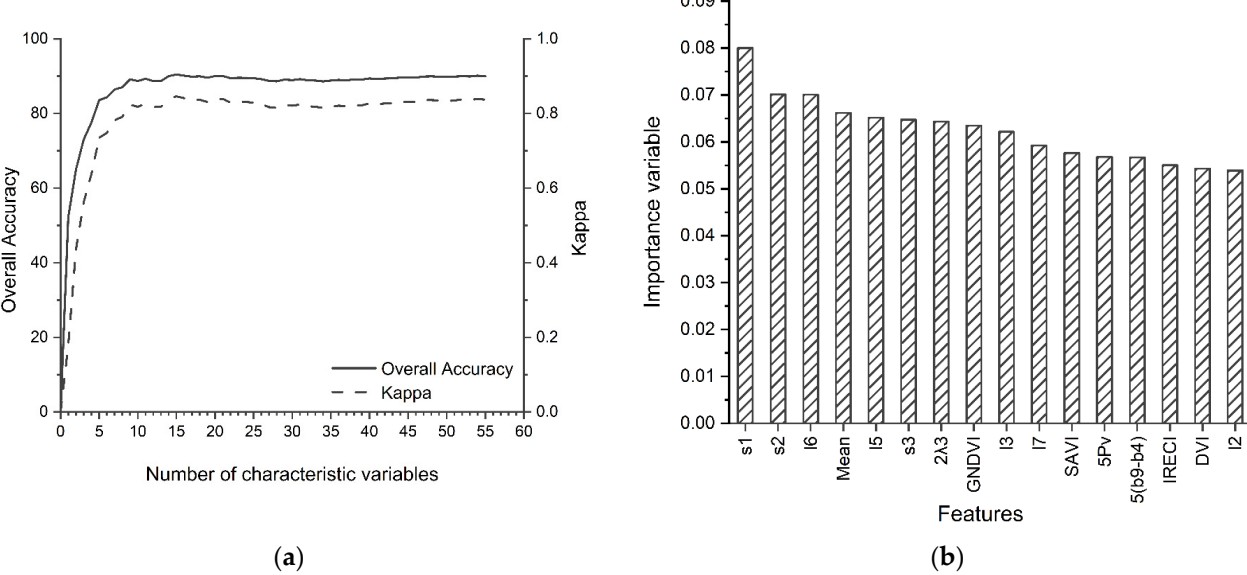

**(a)**                    **(b)**

**Figure 5.** (**a**) The relationship between the number of features and the overall accuracy; (**b**) ranking the importance of preferred features (s1, s2 and s3 are the 1, 2 and 3 bands of Sentinel-2 satellite data, and l2, l3, l5, l6 and l7 are the 2, 3, 5, 6 and 7 bands of Landsat-9 satellite data).

The importance of the variables in the feature set is shown in Figure 5b. The importance of the features is shown as spectral band > texture feature > polarization decomposition parameter > vegetation index. The band importance of Sentinel-2 data is R443, R490, and R560 from high to low. Among them, R443 and R490 belong to the blue wave band, and R560 is the green light band. Chlorophyll strongly reflects green light and strongly absorbs the blue light band. Chlorophyll content in leaves of different mangrove species is different, so there are differences in the intensity of reflection and absorption of green light and blue light, which may be the reason why green light band and blue light bands become the most important to distinguish mangrove species. For Landsat-9 data, the features are R1609, R864.6, R561.3, R2201, and R482.6 in descending order of importance, i.e., short infrared band 1, near infrared band, green light band, short infrared band 2, and blue light band. Previous studies have also confirmed that the short infrared band and the near infrared band are the more important bands for distinguishing mangrove species [40]. In short infrared and near infrared bands, Landsat-9 satellite data have narrower band width than Sentinel-2B satellite data, so Landasat-9 spectral resolution is higher. Therefore, short infrared and near infrared bands of Landsat-9 satellite data are more suitable for mangrove species classification than Sentinel-2B satellite data. The mean importance of texture characteristics was ranked fourth. The texture of AM is finer and more regular, that of SA and KO is rougher and more irregular, and the mean value of SA texture is significantly higher than that of the other two species. GNDVI is the most promising indicator for distinguishing mangrove species, followed by SAVI, IRECI, and DVI. The GNDVI is a vegetation index calculated from a combination of NIR and green light bands. This further proves that near-infrared and green wavelengths can be important features to distinguish mangrove species. Among SAR data characteristics, the L3 characteristic value of coherence matrix $T$ of GaoFen-3 data in February is the most important variable for distinguishing mangrove species in SAR data, ranking seventh in the preferred feature set. The radar signal incident on the mangrove canopy will have significant depolarization. The cross-polarization receiving antenna detects a small part of the depolarization radar energy whose polarization direction changes exactly 90 degrees, and the λ3 eigenvalue corresponds to the square of the cross-polarization component. Therefore, the λ3 eigenvalue increases the difference of the original cross-polarization component. The volume scattering component of SAR data in May ranked the 12th place in the preferred feature subset. The volume scattering model is a scattering model for forest canopy structure. However, the volume and shape of the canopy varies greatly among species. The SA belongs to tall trees, while the KO and AM belong to small shrubs. Therefore, the importance of the volume scattering component is high. In general, the optical data importance is higher than that of SAR data. However, because SAR data can reflect the scattering pattern of canopy structure, it can effectively compensate for the phenomenon of foreign matter co-spectrum in spectral data.

*3.4. Classification Results of Different Methods Based on the Preferred Feature Set*

In order to verify the applicability of the ERT classifier for interspecific classification of mangroves, RF, KNN, and Bayes classifier were selected as the comparison methods. For the control variables, the experimental input features were all preferred feature subsets, and the training and validation sample inputs were the same. Table 6 summarizes the mapping accuracy, user accuracy, overall accuracy and Kappa of the three comparison methods. The classification accuracy of the RF algorithm was 88.47%, Kappa was 0.81, the classification accuracy of the KNN algorithm was 70.74% and the Kappa value was 0.52. The Bayes algorithm had the lowest classification accuracy of 61.62% and the Kappa was 0.42. The highest classification accuracy of the four compared methods was the ERT algorithm, and the classification accuracy of the ERT classification algorithm in this paper was higher than that of the RF classification algorithm, which improved by 1.66%. In order to evaluate the importance of this minor improvement, this paper uses the training data set to perform 10-fold cross-validation on the extreme random tree and random forest algorithms, and

uses the t-test to statistically analyze whether there is a significant difference between the overall accuracy and Kappa coefficient of the two algorithms. The average value of the overall accuracy of the ERT algorithm is 92.54%, the variance is 0.3183, and its Kappa mean value is 0.88 and the variance is 0.00008239. The average value of the overall accuracy of the RF algorithm is 91.24%, the variance is 0.3125, and its Kappa mean is 0.86 and variance is 0.00007550. The $t$ value of the overall accuracy is 5.1762, and the $t$ value of Kappa is 5.2575. According to the table, 5.1762 and 5.2575 are both greater than 1.7341, so it is believed that there is significant difference between ERT and RF. Therefore, the ERT algorithm in this paper was applicable to mangrove species classification and achieved more satisfactory results. In addition, this paper also establishes a spatial transfer matrix analysis for the classification results and sample data of the two methods of random forest and extreme random tree. The spatial transfer matrix analysis table is shown in Tables 7 and 8. The horizontal axis represents the species area of the classification map, the vertical axis represents the species area of the sample map, and the number on the diagonal is the area of the overlap of the classification map and the sample map, that is, the area of the correct classification. After calculation, the area of the extreme random tree classification is greater than that of the random forest, so from another point of view, it is also proved that the classification accuracy of extreme random tree algorithm is better than that of random forest.

**Table 6.** Classification accuracy table of three comparison methods based on preferred features (classification accuracy includes overall accuracy, Kappa, PA, and UA).

| Classifier | | RF | | KNN | | Bayes | |
|---|---|---|---|---|---|---|---|
| | | PA | UA | PA | UA | PA | UA |
| | SA | 0.88 | 0.91 | 0.72 | 0.77 | 0.8 | 0.51 |
| | KO | 0.89 | 0.84 | 0.67 | 0.63 | 0.47 | 0.81 |
| | AM | 0.89 | 0.88 | 0.72 | 0.65 | 0.63 | 0.61 |
| OA | | 88.47% | | 70.74% | | 61.26% | |
| Kappa | | 0.81 | | 0.52 | | 0.42 | |

**Table 7.** Space transition matrix of ERT algorithm (Area unit: ha).

| Sample True Value | Classification Result | | | |
|---|---|---|---|---|
| | SA | AM | KO | Total |
| **SA** | 87.01961569 | 6.293972504 | 6.029115196 | 99.34270339 |
| **AM** | 5.690461859 | 15.28142336 | 0.742151633 | 21.71403685 |
| **KO** | 9.016527601 | 1.381555296 | 23.62586764 | 34.02395053 |
| **total** | 101.7266051 | 22.95695116 | 30.39713447 | 155.0806908 |

**Table 8.** Spatial transfer matrix of random forest algorithm (Area unit: ha).

| Sample True Value | Classification Result | | | |
|---|---|---|---|---|
| | SA | AM | KO | Total |
| **SA** | 86.77964846 | 6.562051645 | 6.000983684 | 99.34268379 |
| **AM** | 5.515079054 | 15.48237008 | 0.716586494 | 21.71403562 |
| **KO** | 8.979577959 | 1.43338697 | 23.61098254 | 34.02394747 |
| **total** | 101.2743055 | 23.47780869 | 30.32855272 | 155.0806669 |

The producer and user accuracies of the RF algorithm were higher than those of KNN and Bayes, indicating that the RF algorithm was more accurate in estimating the three species. The KNN algorithm had higher producer and user accuracies for SA than the other two species, thus estimating SA more accurately than KO and AM. The Bayes classification algorithm had the lowest producer accuracy of 0.47 for KO, which shows that the other two

species were misclassified into KO in a larger area, and the lowest user accuracy of 0.61 for SA, which shows that SA was misclassified into the other two species in a larger area.

We have made comparative experiments with three algorithms of RF, KNN and Bayes, and the ERT algorithm (Figure 6). The overall accuracy and Kappa of the four algorithms are shown in Table 9. The classification accuracy of RF, KNN and Bayes algorithms based on 10 groups of experimental features is lower than the ERT algorithm. The classification accuracy of the ERT algorithm in the optimal feature set is also the highest.

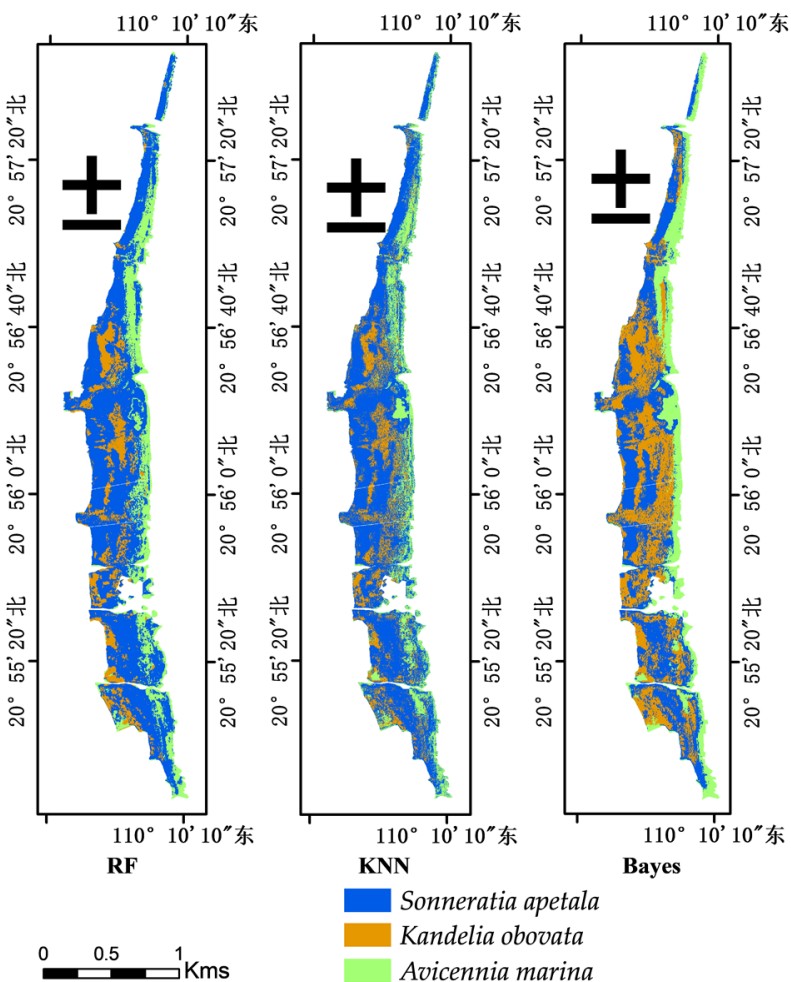

**Figure 6.** Results of different methods of classification.

**Table 9.** Total accuracy and Kappa coefficient of 10 groups of experiments based on 4 methods.

| | | 1 | 2 | 3 | 4 | 5 | 6 | 7 | 8 | 9 | 10 |
|---|---|---|---|---|---|---|---|---|---|---|---|
| **ERT** | OA | 66.59% | 84.91% | 59.68% | 80.45% | 85.49% | 85.10% | 87.95% | 88.51% | 90.00% | 90.13% |
| | Kappa | 0.45 | 0.75 | 0.34 | 0.68 | 0.76 | 0.75 | 0.80 | 0.81 | 0.83 | 0.84 |
| **RF** | OA | 64.35% | 83.72% | 59.94% | 79.35% | 84.36% | 84.43% | 87.28% | 87.37% | 88.91% | 88.47% |
| | Kappa | 0.42 | 0.73 | 0.34 | 0.67 | 0.75 | 0.75 | 0.79 | 0.79 | 0.82 | 0.81 |
| **KNN** | OA | 57.32% | 77.36% | 59.68% | 77.19% | 81.01% | 75.38% | 75.39% | 75.38% | 75.38% | 70.74% |
| | Kappa | 0.28 | 0.63 | 0.32 | 0.63 | 0.69 | 0.60 | 0.60 | 0.60 | 0.60 | 0.52 |
| **Bayes** | OA | 57.32% | 55.42% | 46.42% | 57.72% | 58.12% | 58.57% | 59.51% | 58.09% | 59.05% | 61.26% |
| | Kappa | 0.28 | 0.32 | 0.22 | 0.35 | 0.38 | 0.38 | 0.40 | 0.38 | 0.39 | 0.42 |

## 4. Discussion

### 4.1. Effect of Spatial Resolution of Spectral Data on Species Classification

For spectral data, a large number of studies have shown that the difference in spectral characteristics of different mangrove species is the fundamental reason for interspecific classification of mangroves, but the spatial resolution of spectral data is also the main reason for the accuracy of feature classification [13,41,42]. The lower the spatial resolution, the larger the range of ground objects contained in a pixel; that is, there may be more types of ground objects in one pixel, and the morphological structure of different species is also very different. The canopy area of SA in this paper is significantly larger than that of AM and KO [43]. The canopy area of SA is about 7~14 m$^2$, the canopy area of AM is about 2~8 m$^2$, and the canopy area of KO is about 2~5 m$^2$. If the spatial resolution of spectral data is significantly larger than that of mangrove species, it will lead to a decrease in interspecific classification accuracy, and the classification accuracy of 2 m resolution is 18.32% higher than that of Sentinel-2 data with 10 m resolution, and 2 m resolution is 20.77% higher than that of Landsat-9 data with 15 m resolution. The classification accuracy was improved by 20.77%, but due to the high cost of acquiring high spatial resolution data, subsequent studies can find the most suitable spatial resolution for interspecific classification by actually measuring the canopy area of different mangrove species.

### 4.2. Influence of Spectral Bands on Species Classification in Multispectral Data

The Gaofen-1 satellite data only contain four bands and lack rich band information. The Sentinel-2 data contain 12 bands and the Landsat-9 data contain 7 bands. Therefore, the panchromatic band of Gaofen-1 data is fused with Sentinel-2 and Landsat-9 data, respectively. The fused data has both high spatial and spectral resolution. The rich band information can effectively distinguish mangrove species. The accuracy of mangrove species differentiation using high resolution Sentinel-2 or Landsat-9 satellite data alone has reached more than 80%. The dark blue and blue bands of Sentinel-2 are the most important for the classification of mangrove species rather than short-wave infrared or near-infrared. The possible reason is that there are differences in the chlorophyll content of different species. The appearance of white bone soil is gray-green, the appearance of autumn eggplant is bright green, and the appearance of acanthopanax is dark green. Therefore, these differences lead to different chlorophyll content, resulting in large differences in the reflectivity of different species in the blue light band.

### 4.3. Contribution of SAR Data Features

A previous study showed that the combination of optical and dual-polarized SAR data can improve the accuracy of mangrove interspecific classification [16]. It was also found that the combination of full-polarimetric SAR data with optical data can improve the accuracy of probability mapping of mangrove tree species [43]. However, few studies have investigated the importance of different features of full-polarimetric SAR data and the impact of different SAR data features in different periods on interspecific classification. In this paper, fully polarized Gaofen-3 data were collected for February 2021 and May 2022, and the probability of changes in species distribution during this period was not significant enough to ignore the effect of time. Due to the special habitat of mangroves, their scattering mechanism is also influenced by the tide. SAR data in February were taken at high tide and SAR data in May were taken at low tide. It can also be seen from Figure 7a that the backscattering coefficient value in February is lower than that in May because the water content of mangrove vegetation under the bedding surface kept changing during high tide, the dielectric constant of the bedding surface kept increasing, and the proportion of specular scattering increased, which caused the radar echo signal. Therefore, for the classification of vegetation growing in the intertidal zone, the influence of tide on the backscattering coefficient must be considered.

In this paper, we selected winter and spring SAR data features for mangrove species classification. Some studies showed that backscatter coefficients of different seasons are

also sensitive to mangrove species identification [19]. However, the experimental data used for classification by previous authors are unipolarized data, and this paper uses fully polarized SAR data from different periods. This study does not contain backward scattering coefficient eigenvalues in the subset of preferred features, indicating that backward scattering coefficient features are not always the most important features for mangrove species classification, and different mangrove species are more sensitive to the derived features of SAR data. However, not all polarization decomposition features are important for mangrove species differentiation, which depends on the geometrical characteristics of mangrove species. This paper finds that the third eigenvalue of the polarization coherence matrix in February and the volume scattering component decomposed by Freeman in May are very important for the classification of three species, namely, candel, acanthopanax, and argillaceous soil. As can be seen from Figure 7b, the polarization decomposition characteristic values of different species in different seasons are different. In the polarization decomposition parameters of May, only λ3 is lower than that of February, while other polarization decomposition parameters of May are higher than that of February. Therefore, further study is needed on the intrinsic relationship between the polarization decomposition characteristics of different seasons and the species structure. In this paper, only two periods of SAR data features are used to evaluate their impact on the classification accuracy of mangrove species. Therefore, it is necessary to conduct a more robust analysis of SAR data features of multiple periods in subsequent experiments. The main purpose of this paper is to study the accuracy of predicting mangrove species with the combination of fully polarized SAR data in different periods and multi-spectral characteristics. However, the effects of optical data in different seasons on the accuracy of mangrove species classification are unknown. Therefore, the best month for identifying mangrove species with optical data should be sought in the future. Thus, mangrove species can be identified by combining SAR data at the best time with optical data at the best time.

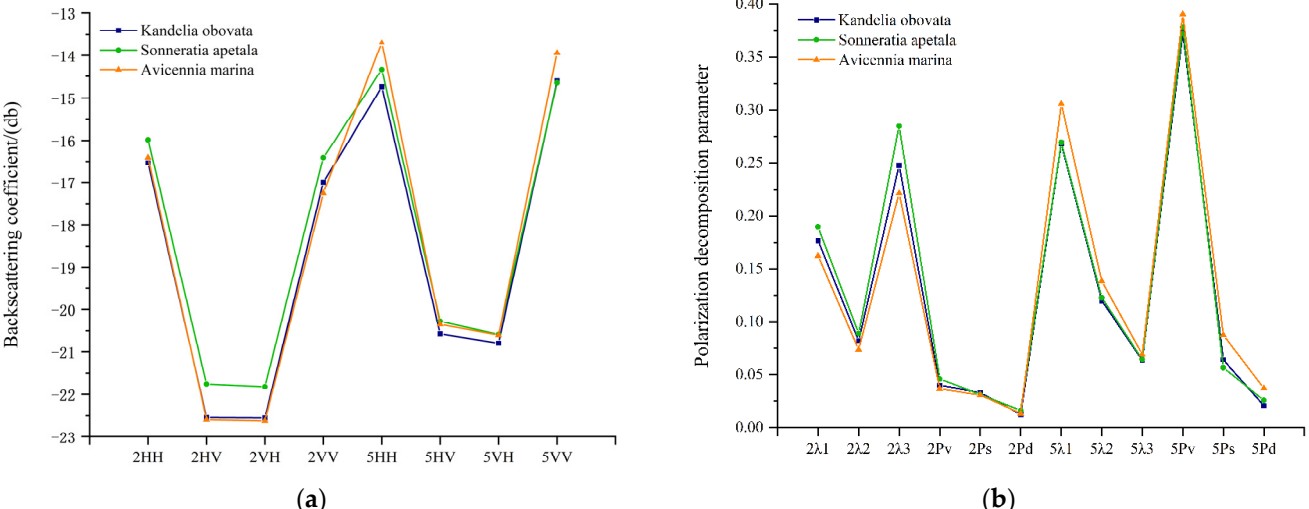

**Figure 7.** (**a**) Backscattering coefficients of different species in February and May; (**b**) polarization decomposition characteristics of different species in February and May.

## 5. Conclusions

In this paper, a mangrove forest in Fucheng town of Zhanjiang City was selected as the research area. For the first time, fusion images of Gaofen-1 data with Sentinel-2 and Landsat-9 data were used for the study of mangrove interspecific classification. Moreover, it was the first attempt to verify the effectiveness of the ERT classification algorithm. A total of 55 features were extracted from Sentinel-2, Landing-9 and GF-3 data of different periods Through the recursive feature elimination algorithm based on the ERT, the optimal feature subset is screened out, and the fundamental reasons why different features are

suitable for the classification of mangrove species are analyzed. In general, the importance of optical data features is higher than that of polarization decomposition features. The deep blue band, blue wave segment, and green band of Sentinel-2 data and the short infrared 1, short infrared 2, near infrared, green band, and blue wave segment of Landsat-9 are the most efficient bands to distinguish mangrove species. The study found that the classification accuracy of Sentinel-2 data with a spatial resolution of 2 m is significantly improved compared with the Sentinel-2 data with a resolution of 10 m. The coupled Sentinel-2 and Landsat-9 data with 2 m resolution have higher classification accuracy than the single data source. In the SAR data, the $\lambda 3$ eigenvalue of the coherence matrix $T$ of GaoFen-3 data in February was the index with most potential for distinguishing mangrove species, followed by the volume scattering component of the data in May. The results show that the polarization decomposition feature is helpful to improve the accuracy of mangrove interspecific classification. In the optimal feature sub-set, RF, KNN, Bayes classification algorithms were chosen to compare with the ERT classification algorithm. The results showed that the classification accuracy of the ERT algorithm was higher than the other three classification algorithms, with an overall accuracy of 90.13% and Kappa of 0.84.

**Author Contributions:** Conceptualization and design of experiments, L.T. and J.F.; conducting experiments, L.T.; Analyzed data and results, L.T. and J.F.; writing—original draft preparation, L.T.; revision and editing of papers, X.W., L.T. and J.F. All authors have read and agreed to the published version of the manuscript.

**Funding:** The work described in the paper was supported in part by the National Natural Science Foundation of China under Grant 42076184, 41876109, 41706195, in part by the National Key Research and Development Program of China under Grant 2017YFC1404902 and Grant 2016YFC1401007, and in part by the National High Resolution Special Research under Grant 41-Y30B12-9001-14/16.

**Data Availability Statement:** Not applicable.

**Acknowledgments:** The authors would like to thank the China National Space Administration (CNSA), United States Geological Survey, and European Space Agency (ESA) for providing the GF, Landsat, and Sentinel datasets, respectively.

**Conflicts of Interest:** The authors declare no conflict of interest.

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
