# Peer review of "Performance Evaluation of Mangrove Species Classification Based on Multi-Source Remote Sensing Data Using Extremely Randomized Trees in Fucheng Town, Leizhou City, Guangdong Province"

_remotesensing, doi:10.3390/rs15051386_

Round 1

Reviewer 1 Report

In general, the manuscript deals with an interesting topic such as the interspecific classification of mangroves. It uses informative tables and graphs that make the text easy to read. However, I have some general comments/suggestions, which I believe can help/improve to enrich the manuscript. I also have specific suggestions in each section which are discussed below.

General comments

-          It is necessary to improve the paragraph where the objective of the work is made clear. As it is currently written between lines 73-86, it does not reflect what was done in the study.

 -          A multi-temporal SAR analysis is discussed in different parts of the manuscript. However, only 2 dates are used, which makes it a simple temporal (not multi-temporal) analysis. I believe this needs to be corrected throughout the text.

 -          The description of the methods should be in the past tense, since everything has already been done and this manuscript is the report of it.

 -          The use of different classification methods and covariates to determine their performance in mangrove species identification requires a more statistically robust evaluation. Currently, I consider that it is not sufficient to use overall accuracy and kappa for this type of articles. There are robust but simple methods such as correlation models and major spatial differences between the classified maps, evaluation models between classifiers, etc. 

-          In addition, I highly recommend performing a spatial uncertainty analysis on the results of the different classifications to the central objective of this study.

Specific comments

Study area (lines 88-99): I suggest including the total study area (ha) in this section.

Line 108:  A total of 15 sample points are mentioned. I suggest including whether these points were selected randomly, systematically, by accessibility, etc.

Lines 120-121: The sentence "…with different spatial resolutions" is repeated.

Line 129: I suggest “…ENVI v5.3”. On line 147 Did you use another version (ENVI 5.6)?

Lines 163-164: Gaofen-2? In Table 1 there are only GaoFen-1 or GaoFen-3.

Line 195: Table 2. Add the description of i and j.

Line 291: It should read, slightly lower than that of group 5?

Line 299: It does mean 87.95% for group 7? it is suggested to rewrite so that it is clear which group it belongs to.

Line 309: total area of mangroves according to which data set used?

Lines 307-314: I suggest including area data for mangrove species from the other classifications. This way a comparison can be made as to which species is the most variable between classifications. This can give us a first idea if there is spatial variation and spectral confusion among the three species studied. Something similar was done for the other 3 classifications in lines 408 to 415.

Tables 4 and 5: Indicate what PA and UA stand for, there is no an explicit reference to them in the main text (producer and user accuracies).

It is suggested to improve the quality of the figures 5(a) and 5(b)

The order of appearance of figure 5 could be changed to match the order in the text of the document. That is, first figure 5(b) would be renamed 5(a) and vice versa.

Figure 5(b): I suggest including a dotted line indicating the optimum number of characteristic variables and kappa.

Figure 5(b): What does mean OA% at left-y-axis?

Lines 365 to 366: Previous studies indicate that infrared bands are the most important for distinguishing mangrove species (Please include references). Then it would be good to include the discussion of why in the case of the sentinel bands the infrared/near infrared bands do not appear as the most relevant (figure 5a) and the blue and green bands appear as the most important.

Lines 389 to 415 (Classification results of different methods based on the preferred feature set): The input features were all preferred subsets (group 10 right?). Why didn't you perform the classification with the all feature combination presented in Table 3? (group 1 to 10). What if one of these combinations and one of the classifier (RF, KNN or Bayes) is better than ERT? I suggest performing the analysis to be sure.

Line 407: “... more satisfactory results”.  I suggest performing a simple statistical test to determine if these differences between classifications (OA and Kappa) are statistically significant.

Lines 438-449: In this paragraph the discussion is based more on spatial resolution than spectral resolution. I suggest to expand the discussion a little more on: a) Line 442 (... and richer band information) What does this mean? and b) The probable reason why the blue bands and the green band came out as the most important in the ERT classification, and not the IR or NIR bands.

Lines 451-454: You mention “Previous studies…”, “…few studies…”  But you only put up one reference. Include those "previous studies" and “few studies” you are pointing out.

Lines 481-483: I believe that it is not possible to speak of the need for a multi-temporal analysis based on an evaluation carried out with only 2 dates. For that it is necessary to perform a much more robust analysis of many dates (multi-temporal). I suggest leaving this idea as a need for further evaluations.

Line 495: Gaofen-2?

Reviewer 2 Report

On a methodological point of view (which is why the paper is submitted) the paper is very interesting with a weakness inherent to this kind of papers: this is a very local and specific case study. How much of what is said (e.g. on the best classifier or the best set of data) would be true for another mangrove? 

The main problem is the redaction.  I am not a native English speaker so I won't judge the level of English but even I detected linguistic problems, starting with a very basic rule of syntax in English a sentence start with capital letter and finishes with a dot ! some paragraphs are unreadable (but easily corrected).

All my comments are added as comments on the pdf file attached. 

If you follow it, the paper can be accepted according to me.

Reviewer 3 Report

1. Need more background information

àHow to overcome the previous study's drawback?: provide info. on proposed method

2. Grammatical error (for example, Field data section)

3. Explanation is unclear for SAR data conversion segment within the optical and SAR data conversion section

4. Description on Extremely randomized trees (ERT) is difficult to understand. Need to input more information about the classification process.

àHow to conduct mangrove interspecies classification?

àHow to prepare training samples?

Round 2

Reviewer 1 Report

  • There are no further comments to the revised version.